

# A multi-scale convolutional neural network-based model for clustering economic risk detection

Yi Zhao

School of Management, Wuhan University of Bioengineering, Wuhan, China

## ABSTRACT

After the public health event of COVID-19, more academics are looking into how to predict combined economic hazards associated with public health incidents. There are currently just a few approaches for detecting aberrant behaviour in aggregated financial risk, and most only work after the economic risk has already been inappropriately aggregated. As a result, we provide a multi-scale convolutional neural network-based model for clustering financial risk anomaly detection (MCNN). First, we use MCNN to train a model for counting economic risks that are used to evaluate aberrant risk aggregating data. Second, we can use the test results to extract the financial risk statistics and economic risk precursor coordinate points. Then, we calculate the economic risk distribution entropy, distance, potential energy and density. To train the three elements of the development state and create the prediction model, we finally use the particle swarm optimization-based extreme learning machine (PSO-ELM). The results of the experiments demonstrate that, in comparison to existing algorithms, our model can efficiently realize early warning and detect abnormal behaviours of aggregated economic risks with high timeliness. Additionally, our method achieves a forecast accuracy of 97.68% and can give additional time to take emergency action.

## INTRODUCTION

Currently, the number of large-scale and ultra-large-scale economic risks is increasing with the development of the social economy, making economic security issues more and more critical. For example, the US subprime crisis in 2008 and the Asian financial crisis were caused by agglomeration behaviour, which resulted from the excessive density of economic risks and eventually caused severe economic accidents. Given this, we conduct research on the aggregation of economic risks to prevent the recurrence of the financial crisis, which is of great significance to ensure the regular operation of the economy (*Liu, 2019*; *Zhao & Zhang, 2021*; *Kaifeng & Li, 2019*).

Recently, the research on economic risk aggregation detection is mainly conducted by the following two aspects. On the one hand, aggregation anomalies are detected by data processing based on economic risk statistics and financial risk density estimation. *Sang, Chen & He (2016)* use distribution entropy and motion speed to propose the detection

Corresponding author
Yi Zhao, zy396187122@163.com

model for economic risk aggregation. *Liu et al. (2019)* proposed an improved social force model, establishing a financial risk behaviour model through economic trajectory clustering and risk interaction force to detect aggregation anomalies. *Zhang et al. (2012)* introduce the concept of social disorder and crowding attributes to construct a group interaction model based on social force through an online fusion strategy for aggregation anomaly detection. The numerical processing methods mentioned above have some defects, such as the lack of estimation of economic risk density and poor timeliness. *Guo et al. (2020)* apply the distributed system to calculate the distribution of financial risks, establish the centre point for the R-number index and use density clustering for aggregation behaviour detection.

Various neural network models have been proposed with the significant progress of convolutional neural networks (CNN) (*Lin et al., 2018*; *He et al., 2017*) in image processing in recent years. In *Yang, Bi & Liu (2018)*, CNN is applied for super-resolution recognition of license plate images. *Shi, Hong & Nan (2018)* employ the selective search method of Faster R-CNN and the target detection model to detect vehicles. At present, the existing CNN can effectively extract features of information. We should regard the aggregation of economic risks as a dynamic procedure. Firstly, we count the financial risks by the improved MCNN to obtain the risk quantity and coordinate information. Then, we compute the three economic risk states: risk density, risk distance potential energy and risk distribution entropy. Subsequently, we feed the three features into the particle sware optimization-extreme learning machine (PSO-ELM) model proposed in this article for training and prediction. At the same time, we can obtain the economic risk state classification model. Finally, we can achieve the prediction results of the aggregation anomalies.

In contrast to the existing methods proposed for the detection of aggregation behaviour, the model in this paper can effectively predict the anomaly of economic risk aggregation. This paper mainly manifests model innovation in the financial risk of abnormal behaviour recognition into a dynamic identification process. We can obtain more accurate coordinates by the density map of the economic risks predicted by MCNN. In addition, we can accurately calculate the density of financial risk, distribution entropy and distance potential energy, reducing the amount of calculation and improving the results compared to the traditional method. The following are the contributions of the proposed approach:

(1) We propose an abnormal aggregation of MCNN-based models for identifying economic risk to identify economic risk base on multi-scale convolutional neural networks.

(2) We train the model to count economic hazards and apply the PSO-ELM to recognize the three economic risk development state components, which can predict abnormal aggregation rather than realizing the detection after the aggregation has formed.

(3) Our method can achieve a forecast accuracy of 97.68% compared to current algorithms and outperform other methods. This shows that our suggested model can effectively implement early warning and detect abnormal behaviours of aggregated economic risks with high timeliness.

## RELATED WORK

Extreme learning machine (ELM) (*Sahu et al., 2022*) is an algorithm proposed by *Huang et al. (2020)* to solve the neural network with a single hidden layer, whose most significant advantage is that its learning speed is much faster than the traditional methods with the premise of ensuring learning accuracy when solving the single hidden layer feedforward neural networks. ELM is a class of machine learning systems or methods based on feedforward neural networks suitable for supervised and unsupervised learning problems. ELM is considered a special type of FNN in research or an improvement on FNN and its backpropagation algorithm. Its characteristics are that the weights of hidden layer nodes are randomly or artificially given and do not need to be updated. The learning process only calculates the output weights. Traditional ELM has a single hidden layer, which is considered advantageous in learning speed and generalization ability compared with other shallow learning systems, such as single-layer perceptron and support vector machine. Some improved versions of ELM have obtained deep structures by introducing self-encoder construction or stacking of hidden layers, enabling representation learning. In this algorithm, the connection weights of the input layer and the hidden layer and the threshold of the hidden layer neurons are generated by the Rand function. In addition, the unique optimal solution can be obtained by setting the number of hidden layer neurons without manual finetuning while training. In this article, we regard the economic risk gathered from the abnormal behaviour as a dynamic process containing three situations: the regular operation, the tendency of abnormal aggregation and the completion of the abnormal collection. Then, we put the features of the three kinds of economic risk status into the PSO-ELM model to train and achieve the prediction results, which can judge the current risk status. The prediction result also includes three types: regular operation, the tendency of abnormal aggregation and the completion of abnormal aggregation.

The predatory behaviour of birds inspires particle swarm optimization (PSO). Massless particles simulate the individual in the birds. PSO is a population-based search process in which each individual is referred to as a particle, defined as a potential solution to a problem optimized in a $D$-dimensional search space, preserving a memory of its optimal historical location and the optimal location of all particles as well as speed. In each evolutionary generation, particle information is combined with adjusting the velocity component on each dimension, which is then used to calculate new particle positions. Particles constantly change their state in a multidimensional search space until they reach equilibrium or optimal conditions or exceed computational limits. The only connection between the different dimensions of the problem space is introduced through the objective function. The particles have two attributes: speed V and position X, which refer to the rate of searching parameters and the direction of search parameters, respectively. The particle searched for the optimal solution alone in the specified interval and recorded the obtained optimal solution as the current individual extreme value $P_{best}$ to share with other particles and to find out the best individual extreme value as the current global optimal solution of the entire particle swarm $G_{best}$. All particles are compared with the optimal global solution and then adjusted to their V and X. Using PSO to optimize ELM can improve classification

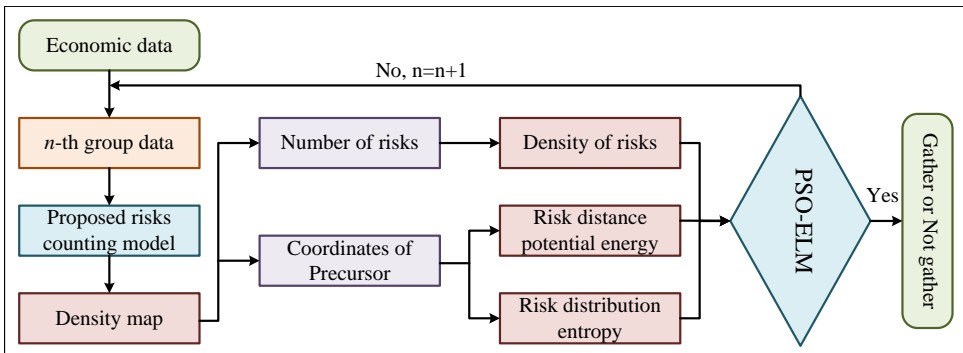

**Figure 1  Flow chart of our method.** The flow chart of the proposed model is shown, where is the number of economic data.

performance. The initial range of speed V used in this article is $(-1, 1)$ and the location X is initialized to $(-1, 1)$. The number of particle swarms is 250.

Principle of the ELM algorithm. (1) Training set: Given Q sets of different samples $(x_i, t_i)$, where $x_i = [x_{i1}, x_{i2}, \ldots, x_{in}]^T$ is the combination of the above features, and $t_i$ refers to a label indicating which group of people the target belongs to. (2) ELM training: W is the connection weight matrix of the input layer and the hidden layer, $\beta$ represents the connection value matrix of the hidden layer and the output layer, b refers to the threshold of the hidden layer neurons and the activation function is g(x). H is the output matrix of the hidden layer of the neural network and $T^T$ is the output of the network. Use the least squares method to solve $\min \|H\beta - T^T\|$ and obtain $\hat{\beta} = H^+$, where $H^+$ is the generalized inverse of H.

## RISKS COUNTING MODEL BASED ON MCNN

The flow chart of the proposed model in this article is shown in Fig. 1, where n is the number of economic data. Currently, the counting methods of economic risks are roughly divided into three categories: (1) Counting based on risk detection, which can obtain the risk counting results by each risk feature in economic data. (2) Counting based on clustering generally applies the Kanada-Lucas-Tomasi-tracking (KLT) and clustering method to estimate the number of economic risks by trajectory clustering. These two methods are only adaptive to the number of economic risks in sparse scenarios and are difficult to count in systems with large risk densities. (3) Regression counting method. Many scholars have conducted in-depth research on regression counting methods, in which Gaussian model regression is one of the typical representatives. However, traditional regression counting still has some defects in feature description and model establishment. In recent years, many scholars have successfully conducted the automatic extraction of practical features by CNN with improved deep learning. We apply the MCNN in this article to better solve the problems of uneven risk density and unobvious risk warning signs.

The advantage of MCNN is that it uses three layers of CNN with different sizes of receptive fields, which can better deal with the difference in the impact of economic risk

precursors caused by the financial data perspective. In addition, the fully connected layer is replaced with the filter convolution layer of $1 \times 1$, making the input data any size and avoiding the loss of economic data information. The network output is the risk density estimation map, from which the total number of financial risks can be obtained.

## Extraction of the density map

The density map has more advantages than directly getting the real economic risks. The density map reflects the distribution of financial risks in economic data. The distribution information is helpful for behaviour analysis because the area with a higher density is more likely to have abnormal economic behaviours. For example, when the model in this article detects aggregation anomalies, the area with a significant risk density can be regarded as a potentially abnormal area. In addition, when the risk density map is used to train the MCNN, the filter can adapt to different risk precursor features, which can make it more suitable for the actual perspective problem and improve the accuracy of the final risk count.

The main reason for using risk precursor features to draw density maps, train the network and carry out risk counting is that risk information is not easy to be disturbed and easily detected in scenarios with large risk densities. Therefore, we apply the way of precursor features for risk counting in dense scenarios.

In this article, we also solve the parameters of the risk density map by applying the geometric adaptive Gaussian kernel and drawing out the risk density map. We introduce the drawing process of the density map as follows. If there are N precursor features in a group of features, the N precursor features can be represented by the following formula:

$$H(x) = \sum_{i=1}^{N} \delta(x - x_i) \tag{1}$$

where we regard the position of the value in the data as x, $x_i$ represents the prominent points of precursor features. $\delta(x - x_i)$ indicates the location of the precursor feature in the data and N is the total number of risks in the group of elements.

The Gaussian kernel function is $G_\delta$. Then the density aggregation function F(x) is obtained as shown in the following formulas:

$$F(x) = \sum_{i=1}^{N} \delta(x - x_i) \times G_{\delta_i}(x) \tag{2}$$

$$\delta_i = \beta \bar{d}^i \tag{3}$$

$$\bar{d}^i = \frac{1}{m} \sum_{j=1}^{m} d_j^i \tag{4}$$

Since it is difficult to obtain the precursor features of the risk in complex scenes, we apply the Gaussian kernel $G_{\delta_i}(x)$ with standard deviation $\delta_i$ and $\delta(x - x_i)$ for convolution to obtain the final risks density F(x). $\bar{d}^i$ is the average distance of the nearest m features to the precursor feature $x_i$. The function can adaptively select $\delta_i$ by the position of each precursor feature. We find that the constant $\beta = 0.3$ can help us to obtain a better risk density map.

## Three features of risk state

The three features of the economic risk state are risk density value, risk distance potential energy and risk distribution entropy. We can achieve the predicted position of the risk precursor feature by the MCNN network, which is slightly different from the real risk feature coordinate and cannot affect the prediction result. We calculate the risk distance potential energy and risk distribution entropy by the obtained risk precursor feature coordinates. We compute the risk density value by the obtained economic risk.

### Computing risk distance energy

Potential energy is a kind of internal energy of the system, a state quantity called potential energy that can be converted into other forms of energy and shared by interacting objects. The potential energy of an object is strongly related to the initial configuration, namely the reference position. In this article, we follow the ideas above, regard the monitoring area as a system and treat the detected risks as objects in the system. Then, we compute each risk's position (coordinates) and calculate the potential energy between risks, namely the distance potential energy.

The risk distance potential energy calculation is mainly determined by calculating the Euclidean distance between individuals. We can achieve only one coordinate for each risk individual obtained by MCNN, much less than the traditional algorithm to calculate the Euclidean distance by multiple feature corners. We calculate the potential energy of risk distance by the following equation:

$$D(x) = \varphi \frac{\sum_{i=1}^{N} \sum_{j=i+1}^{N} C_{ij}}{(N-1)!} \tag{5}$$

where $C_{ij}$ is the Euclidean distance between two coordinates, $\varphi$ refers to the modified value taking a constant and N represents the number of all risks in the economic data.

### Computing risk distribution entropy

Clausius proposes the concept of entropy, which means, in Greek, the change of the intrinsic properties of a system. Then, Boltzmann proposes the statistical physics interpretation of entropy and proves that the macroscopic physical properties of the system can be regarded as the equal probability statistical average of all microscopic states. Thus, entropy can be regarded as a measure of the chaos degree of a system. In modern times, Shannon introduced the concept of entropy in statistical physics into channel communication, initiated the discipline of information theory and proposed information entropy. Information entropy reflects the uncertainty of random events and can measure the amount of information. This article applies the information entropy to describe the risk distribution information. If the risk distribution is discrete, the distribution entropy is large. In contrast, if the risk is aggregated, the distribution entropy is small. Firstly, we normalize the obtained risk coordinates to $[-1, 1]$. Subsequently, we divide $[-1, 1]$ into 20 continuous cells $r_i, i = 1, 2, \ldots, 20$. Finally, we calculated the distribution entropy. The

mathematical expression is as follows:

$$S(k) = -\sum_{i=1}^{20} p_i \log 2p_i \tag{6}$$

$$p_i = \frac{count\,(r_i)}{\sum_{i=1}^{20} count\,(r_i)} \tag{7}$$

where $S(k)$ is the distribution entropy of the first economic data, $p_i$ represents the probability of sample occurrence in the interval and $count(r_i)$ refers to the number of coordinate points in the interval $r_i$ after normalization.

### Computing risk density

When calculating the risk density value, it is essential to obtain the risk in the economic samples accurately. The traditional method uses the Gaussian mixture model to extract the binary foreground pixels of the region of interest. Then corner detection is performed to obtain the feature corners and calculate the corner density by these corners. Finally, a function to fit the risk number is applied in each economic data. However, due to the problem that the risk information is not prominent, the risk statistics of the traditional method are not accurate. In terms of time consumption, the process is cumbersome and time-consuming because of the multiple steps such as foreground detection, foreground pixel normalization, corner detection and function fitting. MCNN does not have these problems. Each economic data can output the total risk number and the risk density map, making it possible to pay special attention to the area with large risk density. Even if it is not prominent, it can also accurately count the risk number. The following equation calculates the specific density calculation:

$$density_{(i)} = \frac{\lambda N_{total(i)}}{S_{area(i)}} \tag{8}$$

where $density_{(i)}$ is the risk density in the i-th economic data, $\lambda$ indicates the correction factor of the total number of risks in the data, $N_{total(i)}$ refers to the total number of people in the i-th economic data and $S_{area(i)}$ is the total amount of economic data. For the convenience of calculation, this article takes 1 for $S_{area(i)}$.

By calculating the risk density of each economic data, we can obtain the change curve of risk density and achieve the information of financial state by the curve.

## EXPERIMENTS

### Dataset

Our method experiments in this article are built on a server with the CPU 8700k, 3.70 GHz, 8.00 GB memory, and graphics card GTX1080-8GB. The MCNN is built, trained and tested in the environment of Anaconda3, CUDA8.0, cudNN8.0.61, PyTorch1.0.1 and Python3.7.3. The calculation of features is completed in Matlab2014a (MathWorks, Inc., Natick, MA, USA). In terms of datasets, we collect 50,000 samples of economic risks containing information about time, location, and scale from the world bank database. We use 30,000 samples as the training set, 10,000 samples as the validation set, and 10,000 as the test set.

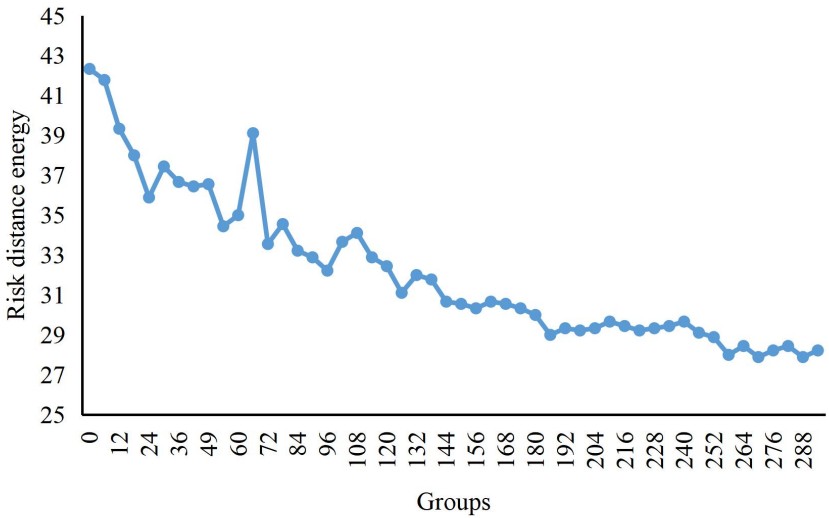

**Figure 2  Potential energy curve of crowd distance.** The change curve of the risk distance potential energy from regular operation to the formation of aggregated anomalies is shown, while the process of economic risks is shown in the graph.

## Features of risk state

We can see the change curve of the risk distance potential energy from regular operation to the formation of aggregated anomalies while the process of economic risks is in the dataset in Fig. 2. Initially, the risk enters the scene from different time points and generates an enormous distance potential energy because there is no risk in the background. As the risks entering from different time points move closer to the central area, the risk distance potential energy begins to decrease. Until the 100th data group, several risks enter the aggregation scene. As the risk data enters, the distance potential energy becomes larger again. By about 124-the data group, no new risks enter, at which point the distance potential energy starts to decrease. Finally, the risk aggregation is completed and the risk distance potential energy remains small.

Figure 3 shows the change curve of risk distribution entropy in the dataset's risk aggregation process. In the beginning, because the economic data from different times enter into the scope of risks, the distribution of entropy value is more extensive, which means that the risk is relatively scattered. However, when the economic data continuously gather in the central area, the entropy distribution decreases slowly. Until the new data after the 100th groups arrive at the scope of the risk, distribution entropy begins to increase. When we compute the 124th data with no new risk data, the distribution entropy begins to decrease until the completion of risk aggregation.

In Fig. 4, we present the change in risk density during the risk aggregation process of the dataset. Initially, only a small number of economic data entered the risk range. As time went by, more and more data entered the risk range, so the risk density has been increasing. From the 150th group to the 160th group, the risk density is relatively stable because no economic data enters the risk range during this period. After the 161st group, as the new

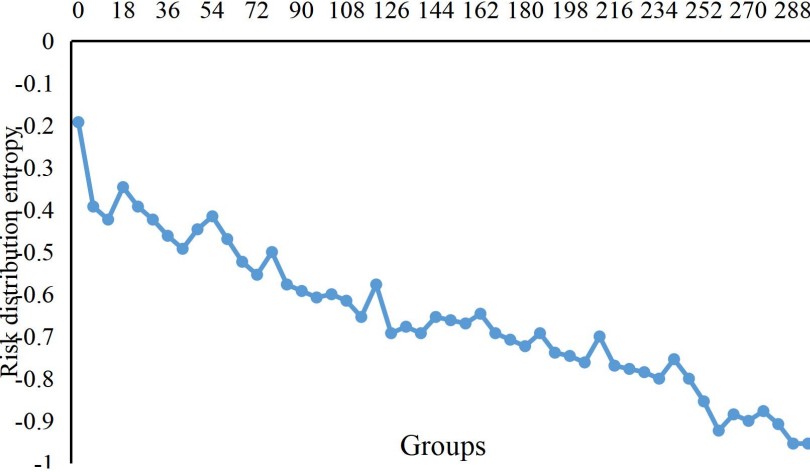

**Figure 3** **Change curve of risks distribution entropy.** The change curve of risk distribution entropy in the dataset's risk aggregation process is shown.

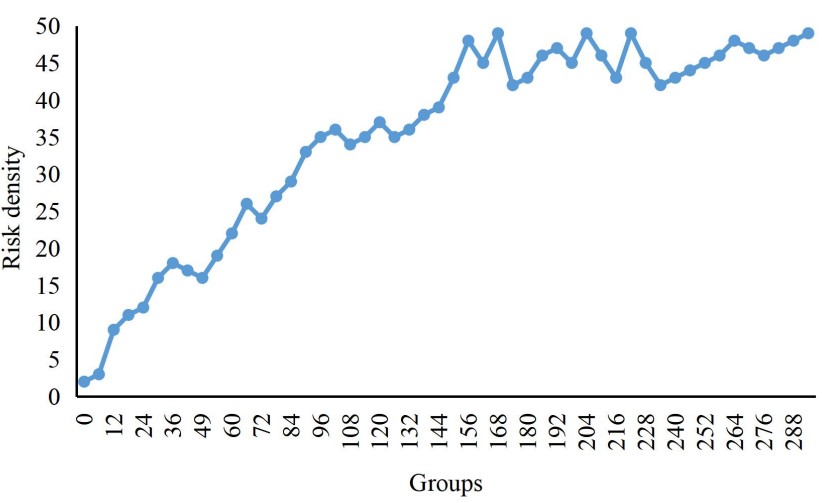

**Figure 4** **Change curve of risks density.** The change in risk density during the risk aggregation process of the dataset is shown.

risk data enters, the density value increases again until no one enters the risk range after 200 frames when the risk density value becomes stable.

## Results and Comparison

In comparison with other studies (*Kou, Peng & Wang, 2014*; *Belhadi, Djenouri & Srivastava, 2021*; *Hu, Gu & Wang, 2019*), the results of our method for the economic risk aggregation recognition are shown in Table 1.

We can see from Table 1 that because the model in this article uses MCNN to solve the problem of non-prominent data perspective and risk features by combining three

**Table 1  Prediction results comparison.** In comparison with other studies (*Kou, Peng & Wang, 2014*; *Belhadi, Djenouri & Srivastava, 2021*; *Hu, Gu & Wang, 2019*), the results of our method for the economic risk aggregation recognition are shown.

| Methods | Groups | | | Accuracy/% | | |
|---|---|---|---|---|---|---|
| | Effective | Normal | Abnormal | Normal | Abnormal | Abnormal state |
| Ours | 756 | 396 | 394 | 98.33 | 97.44 | 97.68 |
| *Liu et al. (2019)* | 268 | 96 | 178 | – | – | 95.66 |
| *Kou, Peng & Wang (2014)* | 222 | 336 | 157 | 94.61 | 97.33 | 92.36 |
| *Belhadi, Djenouri & Srivastava (2021)* | – | – | – | – | – | 93.00 |
| *Hu, Gu & Wang (2019)* | 268 | 96 | 172 | – | – | 90.12 |

risk development state features, the recognition of abnormal aggregation is better than that of other algorithms. In comparison to *Liu et al. (2019)*, our method improves the accuracy by 2.02% because of the fewer and more effective parameters. Compared *Kou, Peng & Wang (2014)* and *Belhadi, Djenouri & Srivastava (2021)*, our approach improves the accuracy of the abnormal state and abnormal samples by over 4%, resulting from less hardware application. In addition, our method outperforms *Hu, Gu & Wang (2019)* in terms of all metrics. In conclusion, our method achieves state-of-the-art performance for risk clustering in the financial industry.

## CONCLUSION

This study suggests an abnormal aggregation of MCNN-based models for identifying economic risk. Studies demonstrate that the model can accurately recognize the condition of economic risk and forecast and identify aberrant financial risk aggregation behaviour. The advantage of the model in this article over the conventional economic risk aggregation identification algorithm is that it predicts abnormal aggregation rather than realizing the detection after the aggregation has formed. This can give more time for early warning of abnormal economic risk aggregation and the corresponding emergency measures because economic risk density and risk aggregation are factors in most financial mishaps. Therefore, we can predict economic risk by our model and formulate corresponding policies in time. All the irregularities in economic risk aggregation have been identified in this research. The issues with the current method of finding aggregation anomalies will be resolved in the future. In future, we will explore research on the relationship between economic risk and aggregation.

### Funding

This work was supported by the Key Project Supported by 2022 Special Program of Hubei Provincial Education an Science Plan "Research on the Current Situation and Breakthrough Path of College Students' Spiritual Dilemma from the Perspective of the New Era" Number: 2022ZA032). The funders had no role in study design, data collection and analysis, decision to publish, or preparation of the manuscript.

## Grant Disclosures
The following grant information was disclosed by the author:
Key Project Supported by 2022 Special Program of Hubei Provincial Education an Science Plan "Research on the Current Situation and Breakthrough Path of College Students' Spiritual Dilemma from the Perspective of the New Era:  2022ZA032.

## Competing Interests
The authors declare there are no competing interests.

## Author Contributions
- Yi Zhao conceived and designed the experiments, performed the experiments, analyzed the data, performed the computation work, prepared figures and/or tables, authored or reviewed drafts of the article, and approved the final draft.

## Data Availability
   The data is available at Kaggle: https://www.kaggle.com/datasets/madhurpant/world-economic-data. The code is available in the Supplemental Files.

## Supplemental Information
Supplemental information for this article can be found online at http://dx.doi.org/10.7717/peerj-cs.1404#supplemental-information.

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
