# Peer review of "A multi-scale convolutional neural network-based model for clustering economic risk detection"

_PeerJ Computer Science, doi:10.7717/peerj-cs.1404_

## Round 0.1 · original submission · Major Revisions

Dear contributors,
Please see below the suggestions of the experts and revise your paper accordingly, and resubmit.

Reviewer 1 ·

Basic reporting

In this paper, the authors propose a cluster economic risk anomaly prediction model based on multi-scale convolutional neural network. They trained an economic risk counting model using the MCNN to test abnormal data for risk aggregation. Secondly, economic risk statistics and economic risk precursor coordinate points were obtained from the test. Then they calculated the economic risk density, economic risk distance potential energy and economic risk distribution entropy. three characteristics of economic risk development state are trained by PSO-ELM to realize the forecast model. In my opinion, the proposed model can realize the early warning and detect the abnormal behavior of aggregate economic risk and has strong timeliness, overall the paper seems to be good, it can be further imporved by the following suggestions incorporation
1. Revise the language on COVID-19 to accurately express the research context under "public health event";
2. Please add 2-3 more keywords, the current 3 keywords are not enough;
3. Supplement the research contribution of this paper (theoretical/practical contribution) at the end of the introduction;
4. There is no good summary of the past work in Section 2 Related works, please add more citations and descriptions;
5. Improve the logic between statements, such as “In recent years, with the significant progress of deep learning in image processing, many scholars have successfully conducted the automatic extraction of practical features by CNN.”
6. Where did the 50,000 economic risk samples used in the experiment come from? More needs to be added about data sets and data processing methods;
7. As for the compared literatures [3,13,14,15], the names of their methods, advantages and other information are introduced in the paper to facilitate better comparison;
8. Paying particular attention to English grammar, spelling, and sentence structure so that the goals and results of the study are clear to the reader.
9. Please make sure your 'conclusion' section underscore the scientific value added of your paper, and/or the applicability of your findings/results, as indicated previously. Please revise your conclusion part into more details.

Experimental design

In this paper, the authors propose a cluster economic risk anomaly prediction model based on multi-scale convolutional neural network. They trained an economic risk counting model using the MCNN to test abnormal data for risk aggregation. Secondly, economic risk statistics and economic risk precursor coordinate points were obtained from the test. Then they calculated the economic risk density, economic risk distance potential energy and economic risk distribution entropy. three characteristics of economic risk development state are trained by PSO-ELM to realize the forecast model. In my opinion, the proposed model can realize the early warning and detect the abnormal behavior of aggregate economic risk and has strong timeliness, overall the paper seems to be good, it can be further imporved by the following suggestions incorporation
1. Revise the language on COVID-19 to accurately express the research context under "public health event";
2. Please add 2-3 more keywords, the current 3 keywords are not enough;
3. Supplement the research contribution of this paper (theoretical/practical contribution) at the end of the introduction;
4. There is no good summary of the past work in Section 2 Related works, please add more citations and descriptions;
5. Improve the logic between statements, such as “In recent years, with the significant progress of deep learning in image processing, many scholars have successfully conducted the automatic extraction of practical features by CNN.”
6. Where did the 50,000 economic risk samples used in the experiment come from? More needs to be added about data sets and data processing methods;
7. As for the compared literatures [3,13,14,15], the names of their methods, advantages and other information are introduced in the paper to facilitate better comparison;
8. Paying particular attention to English grammar, spelling, and sentence structure so that the goals and results of the study are clear to the reader.
9. Please make sure your 'conclusion' section underscore the scientific value added of your paper, and/or the applicability of your findings/results, as indicated previously. Please revise your conclusion part into more details.

Validity of the findings

In this paper, the authors propose a cluster economic risk anomaly prediction model based on multi-scale convolutional neural network. They trained an economic risk counting model using the MCNN to test abnormal data for risk aggregation. Secondly, economic risk statistics and economic risk precursor coordinate points were obtained from the test. Then they calculated the economic risk density, economic risk distance potential energy and economic risk distribution entropy. three characteristics of economic risk development state are trained by PSO-ELM to realize the forecast model. In my opinion, the proposed model can realize the early warning and detect the abnormal behavior of aggregate economic risk and has strong timeliness, overall the paper seems to be good, it can be further imporved by the following suggestions incorporation
1. Revise the language on COVID-19 to accurately express the research context under "public health event";
2. Please add 2-3 more keywords, the current 3 keywords are not enough;
3. Supplement the research contribution of this paper (theoretical/practical contribution) at the end of the introduction;
4. There is no good summary of the past work in Section 2 Related works, please add more citations and descriptions;
5. Improve the logic between statements, such as “In recent years, with the significant progress of deep learning in image processing, many scholars have successfully conducted the automatic extraction of practical features by CNN.”
6. Where did the 50,000 economic risk samples used in the experiment come from? More needs to be added about data sets and data processing methods;
7. As for the compared literatures [3,13,14,15], the names of their methods, advantages and other information are introduced in the paper to facilitate better comparison;
8. Paying particular attention to English grammar, spelling, and sentence structure so that the goals and results of the study are clear to the reader.
9. Please make sure your 'conclusion' section underscore the scientific value added of your paper, and/or the applicability of your findings/results, as indicated previously. Please revise your conclusion part into more details.

Additional comments

In this paper, the authors propose a cluster economic risk anomaly prediction model based on multi-scale convolutional neural network. They trained an economic risk counting model using the MCNN to test abnormal data for risk aggregation. Secondly, economic risk statistics and economic risk precursor coordinate points were obtained from the test. Then they calculated the economic risk density, economic risk distance potential energy and economic risk distribution entropy. three characteristics of economic risk development state are trained by PSO-ELM to realize the forecast model. In my opinion, the proposed model can realize the early warning and detect the abnormal behavior of aggregate economic risk and has strong timeliness, overall the paper seems to be good, it can be further imporved by the following suggestions incorporation
1. Revise the language on COVID-19 to accurately express the research context under "public health event";
2. Please add 2-3 more keywords, the current 3 keywords are not enough;
3. Supplement the research contribution of this paper (theoretical/practical contribution) at the end of the introduction;
4. There is no good summary of the past work in Section 2 Related works, please add more citations and descriptions;
5. Improve the logic between statements, such as “In recent years, with the significant progress of deep learning in image processing, many scholars have successfully conducted the automatic extraction of practical features by CNN.”
6. Where did the 50,000 economic risk samples used in the experiment come from? More needs to be added about data sets and data processing methods;
7. As for the compared literatures [3,13,14,15], the names of their methods, advantages and other information are introduced in the paper to facilitate better comparison;
8. Paying particular attention to English grammar, spelling, and sentence structure so that the goals and results of the study are clear to the reader.
9. Please make sure your 'conclusion' section underscore the scientific value added of your paper, and/or the applicability of your findings/results, as indicated previously. Please revise your conclusion part into more details.

Reviewer 2 ·

Basic reporting

The paper proposes an economic risk identification model that utilizes MCNN to detect and predict abnormal economic risk aggregation. This model offers a reasonable contribution by providing an advantage over traditional identification algorithms, which only detect aggregation after it has formed. By predicting abnormal aggregation, the model allows for early warning and the implementation of emergency measures, which is crucial in preventing financial accidents related to risk concentration and density.

While the paper has innovative aspects, there are some suggestions for improvement that need to be addressed.

1. In general, there is a lack of explanation of replicates and statistical methods used in the study.
2. The flowchart available in Fig. 1 needs a revision as the symbols do not follow the standard characteristics.
2. The introduction of MCNN is inadequate, including its structure, algorithm flow, etc.;
3. It is important to enhance the linguistic expression of the concluding section and add information to it regarding the limitations and potential avenues for future development.

Experimental design

It is necessary to furnish a rationale for conducting the different experiments, explaining why the authors chose to carry them out.

The dataset description is inadequate. Details of preprocessing is necessary to replicate the experiments. Details of model hyper parameters are also not provided.
The process of group construction is also missing.

Validity of the findings

The author should place the findings in the context of existing literature and clearly articulate the unique contributions of their study to the field..

The discussion section should present a cohesive and logical set of arguments that may provide insights into the significance of the authors' proposals.

The evaluation of result must be done in the context of English research works.

Additional comments

The quality of literature review and citation is not up to the mark. Authors should consider contextualize the work with the help of state of the art work in the relevant field.

---

## Round 0.2 · Minor Revisions

Thank you for your revision, However, few minor changes still needed to be done. Please provide the alternate to references in table 1 or give a translation site in your response. Thank you

Reviewer 1 ·

Basic reporting

The paper has been revised well according to my previous comments. therefore, i recommend its acceptance

Experimental design

The experiments are well explained and justified to validate the results.

Validity of the findings

Satisficed with the revised version of the manuscript.

Additional comments

THe paper is revised well. i have no more comments

Reviewer 2 ·

Basic reporting

The authors have revised the manuscript reasonably.

Experimental design

The authors have explained the experimental setup reasonably.

Validity of the findings

references 5, 13, 14 and 15 in Table 1 are all Chinese language based references. It is not possible for non Chinese or Mandarin person to comment on the results or to understand the context of comparison.

---

## Round 0.3 · accepted · Accept

Thanks for providing the alternatives and incorporating the experts' suggestions. Your paper is being recommended for publication.